# Reference ranges and Z-scores for fetal cardiac measurements from two-dimensional echocardiography in Asian population

Eric C. Lussier[1]☯, Shu-Jen Yeh[2,3], Wan-Ling Chih[1,2]☯, Shan-Miao Lin[2,3], Yu-Ching Chou[1], Szu-Ping Huang[1], Ming-Ren Chen[2,3,4]*, Tung-Yao Chang[1]

**1** Taiji Clinic, Taipei, Taiwan, **2** Department of Pediatrics, MacKay Memorial Hospital, Taipei, Taiwan, **3** Mackay Medical College, Taipei, Taiwan, **4** Mackay Junior College of Medicine, Nursing, and Management, Taipei, Taiwan

☯ These authors contributed equally to this work.
* mingren44@gmail.com

**Data Availability Statement:** All relevant data are within the manuscript and Supporting Information files.

## Abstract

Currently available fetal echocardiographic reference values are derived mainly from North American and European population studies, and there is a lack of reference z-score for fetal echocardiographic measurement in Asian populations. The aim of this study was to establish normal ranges of echocardiographic measurements and z-scores in healthy Asian fetuses. A total of 575 healthy pregnant Taiwanese with an estimated gestational age from 14 to 38 weeks were enrolled voluntarily for this observational study. Standard two-dimensional echocardiography was performed to obtain measurements of the cardiac chambers and great arteries of the developing fetuses. In contrast to past studies, our sample was more evenly distributed for estimated gestational age (p<0.001). We present percentile graphs for 13 fetal echocardiographic measurements from the knowledge of estimated gestational age, biparietal distance, head circumference, abdominal circumference, and femur length. Most cardiac structures and developmental markers had linear models as the best-fitting, except for transverse aortic isthmus by estimated gestational age and transverse ductus arteriosus by femur length. Our findings indicate that estimated gestational age was generally the best model for fetal heart development, while head circumferences could be used as an optimal developmental marker to predict left atrium, right atrium, right ventricle, pulmonary annulus, and ductus arteriosus. Lastly, we developed nomograms for each of the 13 fetal heart measurements by each developmental markers. This is the first study providing echocardiographic reference ranges and nomograms for Asian fetuses. Computing z-scores from nomograms helps in standardizing comparisons and adds additional prognostic information to the diagnosis of congenital heart disease.

## Introduction

Two-dimensional (2D) echocardiography is currently one of the most practical noninvasive methods to measure cardiac structures for fetuses prenatally and children postnatally. Reference values and Z-scores for fetal cardiac dimensions derived from 2D echocardiography are

**Funding:** The author(s) received no specific funding for this work.

**Competing interests:** The authors have declared that no competing interests exist.

well-established [1–12], allowing quantification and comparison of size of cardiac structures in differing subgroups of a disease [13]. In 1990s, several studies on fetal cardiac measurements using B-mode ultrasonography were published, providing regression equations and 95% confidence intervals based on gestational age [1–3]. In 2005, Schneider *et al.* reported reference ranges as well as z-scores, not only based on gestational age, but also based on non-cardiac fetal biometric parameters (biparietal diameter and femur length) [6]. The computation of z-scores provides more information than just normality, allowing more precise evaluation of the cardiac structure when the measurement is below or above 95% confidence intervals.

In clinical practice, z-scores references are practical not only in the screening and diagnosis of fetal cardiac structural abnormalities [14–18], but fetal cardiologist also use z-scores to predict and counsel about possible postnatal outcome and treatment strategies [19–22]. However, currently available z-score calculators are based on studies from Caucasian populations. Fetal echocardiographic reference values for the Chinese population had been published, but z-scores were not provided [11]. Z-score reference range for normal fetal heart size have been reported in Asian population, but not for specific cardiac structures [9]. Our aim was to construct normal ranges and z-scores for fetal cardiac structures, in the 14–38 weeks of gestational period among a Sino-origin population sample.

## Materials & methods

A total of 599 healthy pregnant Taiwanese mothers with an estimated gestational age (EGA) from 14 to 38 weeks were enrolled from September 2016 until December 2017. Cases received measurements prospectively at 3 clinics in northern Taiwan from an unselected population. We recruited only women with singleton pregnancies and regular menstruation, and had a measurement of the crown-rump length that confirmed EGA. We include only fetuses without growth restriction based on fetal biometry of the Taiwanese fetuses [23].

A total of 24 fetuses were found to be abnormal and excluded. Fetuses were retrospectively excluded if there were any maternal disease diagnosed during the pregnancy or any structural abnormality diagnosed either prenatally or postnatally. Exclusion criterions for abnormality included: small-or large-for-gestational age, nuchal translucency greater than the 95th centile at 11–14 weeks, or any chromosomal/genetic abnormalities. Each subject was studied cross-sectionally in order to avoid potential collinearity bias of including serial measurements of the same fetus. The study was approved by institutional review board of Mackay memorial hospital (16MMHIS041e 20160300003). An informed consent was obtained in written format from every participant before enrollment.

### Instrumentation

Fetal measurements were performed using ProSound Alpha 6 (Hitachi, Tokyo, Japan) and ProSound F75 (Hitachi, Tokyo, Japan). All pregnancies were examined transabdominally with 5.0-MHz probes in the 14–38 week period. Images were recorded digitally and stored securely.

### Echocardiography and measurements

All fetal examinations were performed by an experienced examiner (Szu-Ping Huang), and reviewed by an obstetrician-gynecologist and a pediatric cardiologist. No intra-observer variability was performed. Measurements of fetal heart structures and developmental markers were done according to guidelines for standard imaging planes from the American Society of Echocardiography [24]. All measurements were reported in centimeters, with the exception of HA which used centimeters$^2$. Heart length (HL), heart width (HW), heart circumference (HtC), heart area (HA), chest circumference (CC) and chamber width were assessed in the

four-chamber view in end-diastole with closed atrioventricular valves. HL was measured from base to apex, while HW was measured at the level of the atrioventricular valve. HtC and HA were measured by tracing along the outer border of the heart. CC was measured using ellipse covering the outer borders of the ribs. Width of left atrium (LA), right atrium (RA), left ventricle (LV) and right ventricle (RV) were measured just above or below the atrioventricular valve orifice, at the level where the diameter was largest and when maximal dilatation occurred in end-diastole. In LVOT and RVOT views, diameter of aortic annulus (Ao) and pulmonary annulus (PA) were measured at the level of the valve in diastole (when the valve is closed). In three-vessel-trachea view, we measure transverse aortic isthmus (AI) diameter and transverse ductus arteriosus (DA) diameter at its junction into each other when widest systolic diameter occured. All measurements were made from inner edge to inner edge. Fetal developmental markers including: biparietal diameter (BPD), head circumferences (HdC), abdominal circumference (AC), and femur length (FL) were concurrently measured during the same visit.

## Grouping and data management

EGA was binned into 2-week intervals from 14 weeks to 38 weeks gestational age. Thus, a fetus that had received a cardiac measurement at 21 weeks and 6 days would fall in the 21 weeks and 4 days to 23 weeks and 3 days interval and would be grouped in the 22-week gestational age group. Other developmental markers were also binned and derived by ensuring normality of distribution between intervals, as well as optimization of representation in each category. The binned groupings were as follows: bi-parietal distance (BPD) (<4.5, 4.5–5.4, 5.5–6.4, 6.5–7.4, 7.5–8.4, ≥8.5), femur length (FL) (<3.5, 3.5–4.4, 4.5–5.4, 5.5–6.4, ≥6.5), abdominal circumference (AC) (<13, 13.0–14.9, 15.0–16.9, 17.0–18.9, 19.0–20.9, 21.0–22.9, 23.0–24.9, 25.0–26.9, 27.0–28.9, 29.0–30.9, ≥31.0), and head circumference (HdC) (<15.0, 15.0–16.9, 17.0–18.9, 19.0–20.9, 21.0–22.9, 23.0–24.9, 25.0–26.9, 27.0–28.9, 29.0–30.9, ≥31.0). A Kolmogorov-Smirnov test of normality was conducted to assess normality of distribution in each developmental marker binned group throughout the developmental timeline. If the normality assumption was found to be violated in more than one group, transformations of the cardiac measurement variables was performed to return the distribution to normality. Transformation order was selected based on the Krishnan et al. (2016) paper ($y^2$, $y^3$, $\ln(y)$, $\sqrt{(y)}$, $1/y$, $1/y^2$, $1/\sqrt{(y)}$, $1/y^2$). If the transformations did not improve the normality of the distribution in each group, higher order equations were used to ensure normality was attained.

In order to construct nomograms, fetal heart structure measurements were binned for normality of distribution between binned groupings and for optimization of representation within groups. For simplicity, range notation upper limited was always rounded down. For example, as "5.0 ± 1.0" which denoted a range of 4.0–5.99cm. Whereas "0.1 ± 0.05" would signify a range from 0.05–0.149cm. For heart circumference (HtC) measurements were categorized into 8 groups (<4.0, 5.0±1.0, 7.0±1.0, 9.0±1.0, 11.0±1.0, 13±1.0, 15±1.0, ≥16.0). Other fetal heart structure binned categorizations can be found in the supplementary figures.

## Statistics

In order to illustrate overall distribution of cases throughout the gestational age, we compared our sample distribution to past studies along the gestational age range. Our sample was compared to two studies done by Shapiro et al. (1998) [3] and Krishnan et al. (2016) [10] by case distribution because both represent important studies on fetal heart biometry that had used similar parameters and markers as our study. A 2-sample Kolmogorov-Smirnov test was employed to compare if the distributions were significantly different in distribution.

Best fitting equations were obtained by use of best-fit model selection method. Linear, quadratic and cubic models were tested and selected by the following criteria: minimizing Akaike's Information Criteria (AIC) and root mean squared error (RMSE). Adjusted R-squared values allowed for comparisons between developmental marker models for each fetal cardiac structure. Furthermore, centile graphs for each fetal heart measurement by each developmental marker (EGA, BPD, FL, AC, and HdC) were provided. Mean regression lines, as well as the 95% CI (2.5[th] and 97.5[th] percentile lines) were plotted and compared by heart structures for each developmental marker.

Lastly, nomograms were developed for all 13 fetal heart structures and each developmental marker. Nomograms are a helpful tool to establish z-score when developmental markers and fetal heart measurement are obtained. To construct the nomograms, a method developed by Schneider et al. was followed (2005). All measurements were transformed with by natural log transformation, as indicated by previous paper. Z-scores were obtained using the following formula:

$$Z - score = (\ln(actual) - \ln(predicted))/root\ MSE$$

Z-scores were obtained by stratifying by developmental markers. The z-scores were then plotted using the XLStat package's scatter plot with regression lines function. All other statistical analyses were performed using SPSS V22.0.

## Results

A total 575 normal healthy fetuses were included in our sample. The sample distribution was compared to the sample distribution in past studies by Shapiro et al. [3] (Fig 1a) and Krishnan et al. [10] (Fig 1b) for each EGA group from 14–38. Shapiro et al. had more cases in earlier

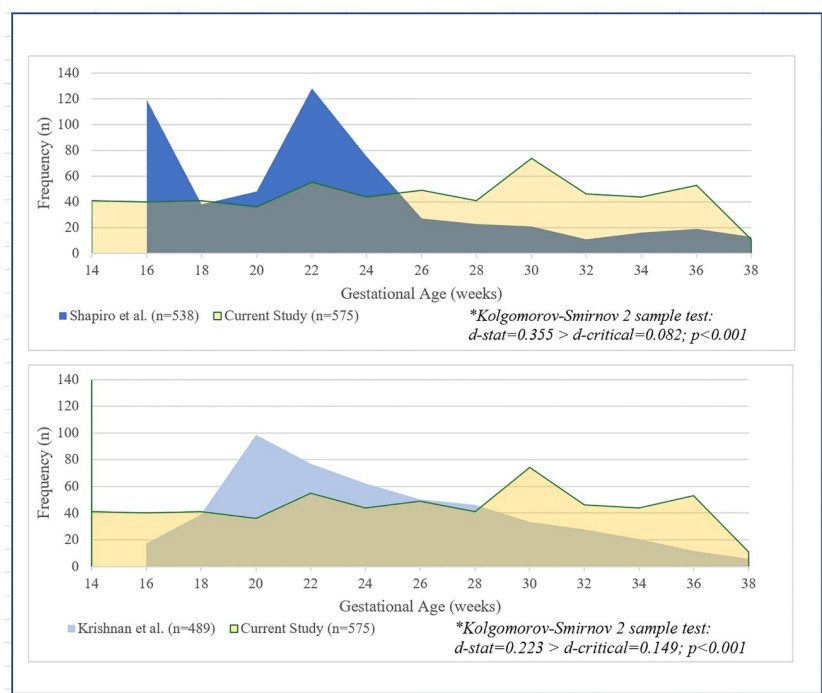

**Fig 1. a, b.** Distribution of cases compared to past studies with normal ranges for heart structures.

EGA pregnancies, while our sample had a relatively equal distribution of study subjects based on gestation age. A 2-sample Kolmogorov-Smirnov test of distribution equality showed that we had a significantly different distribution from that of Shapiro et al. *(d-stat = 0.355 > d-critical = 0.082; p<0.001)*, and to the more recent study by Krishnan et al. *(d-stat = 0.223 > d-critical = 0.149; p<0.001)*.

The best fitting equations for the 13 fetal heart structures were reported by each developmental marker (Table 1). A forward best-fitting model was used to determine the optimal model. All model selection resulted in linear models being selected as those that minimized AIC and RMSE, except for the transverse arteries: Transverse Aortic Isthmus by EGA ($AI = -0.18^*EGA^2 + 1.97^*EGA + 0.28$) and Transverse Ductus Arteriosus by FL ($DA = -0.13^*FL^2 + 1.31^*FL + 0.32$). Our findings indicate that EGA was the optimal marker for: HW (adj. $R^2 = 0.928$), HL (adj. $R^2 = 0.939$), HtC (adj. $R^2 = 0.948$), HA (adj. $R^2 = 0.972$), CC (adj. $R^2 = 0.964$), LV (adj. $R^2 = 0.848$), Ao (adj. $R^2 = 0.859$), and AI (Quadratic: adj. $R^2 = 0.749$). On the other hand, HdC was an optimal marker for: ln(LA) (adj. $R^2 = 0.858$), RA (adj. $R^2 = 0.878$), ln(RV) (adj. $R^2$: EGA = estimated gestational age, BPD = bi-parietal distance, FL = femur length, AC = abdominal circumference, HC = head circumference†EGA = estimated gestational age, BPD = bi-parietal distance, FL = femur length, AC = abdominal circumference, HC = head circumference.

Centile graph tracking development of heart circumference by EGA, BPD, FL, AC and HdC were plotted and reported in Fig 2. Centile graphs of other fetal heart structures can be found in Supplementary Materials (S1a–S1m Fig).

Lastly, nomograms were developed for HtC from the knowledge of each developmental marker (Fig 3). The nomograms are useful tools for physicians to quickly assess z-score of a certain heart structure according to developmental marker measurements. Nomograms for all other fetal heart structures can be found in the Supplementary Materials section (S2a–S2m Fig).

## Discussion

We present regression equations, centile graphs and nomograms for 13 fetal echocardiographic measurements from 14 to 38 weeks in Taiwanese sample, allowing calculation of z-scores for these cardiac structures in fetal life from knowledge of EGA, BPD, FL, AC, and HdC. Although reference ranges of fetal cardiac measurements in an Asian population has been published previously [11], our study is the first to provide nomogram representation in an Asian population and with a full range of developmental markers. In addition, the sample selection was collected with even distribution throughout the gestational age. We employed a standardized selection criteria for model selection, which resulted in linear model selection for most structures. Furthermore, estimated gestational age and head circumference were shown to be the best markers for predicting fetal cardiac growth.

A strict inclusion and exclusion criteria ensured that developmental reference ranges were based on normal cases that were normally distributed or transformed if the normality distribution assumption was violated. Fetuses aged 14–38 weeks comprised our sample, with a relatively equal distribution of study subjects based on gestational age. Selection of cardiac structure development were done by comparing linear, quadratic and cubic models. Most structures resulted in a linear model selection. In a review by Devore [25], equality of frequency between different developmental ages was a necessary item for ensuring quality of centile and z-score values derived from the sample. This is a feature of our sample which ensured representativeness of fetal growth throughout pregnancy. In other published studies on reference range of fetal echocardiography [3, 10], data were mainly collected during the second

**Table 1. Best fitting models for each fetal heart structures.**

| Structures[a] | Markers | n | Transformed | Best Fit Model | Best Fitting Equation | Adj.-$R^2$ | AIC | RMSE |
|---|---|---|---|---|---|---|---|---|
| Heart Width | EGA* | 575 | None | Linear | HW = 0.12*EGA—0.76 | 0.928 | -1622.8 | 0.243 |
| | BPD | 508 | None | Linear | HW = 0.45*BPD—0.95 | 0.868 | -1240.4 | 0.294 |
| | FL | 508 | None | Linear | HW = 0.58*FL + 0.94 | 0.856 | -1196.6 | 0.307 |
| | AC | 508 | None | Linear | HW = 0.25*AC + 1.06 | 0.903 | -1397.8 | 0.252 |
| | HdC | 508 | None | Linear | HW = 0.26*HdC + 0.86 | 0.897 | -1365.3 | 0.260 |
| Heart Length | EGA* | 574 | None | Linear | HL = 0.16*EGA—0.96 | 0.939 | -1437.7 | 0.285 |
| | BPD | 507 | None | Linear | HL = 0.57*BPD + 1.23 | 0.878 | -1027.8 | 0.362 |
| | FL | 507 | None | Linear | HL = 0.74*FL—1.22 | 0.870 | -996.1 | 0.374 |
| | AC | 507 | None | Linear | HL = 0.32*AC—1.38 | 0.910 | -1184.6 | 0.310 |
| | HdC | 507 | None | Linear | HL = 0.33*HdC + 1.13 | 0.904 | -1153.4 | 0.320 |
| Heart Circumference | EGA* | 575 | None | Linear | HtC = 0.49*EGA—2.64 | 0.948 | -255.3 | 0.800 |
| | BPD | 508 | None | Linear | HtC = 1.73*BPD -1.07 | 0.891 | 19.1 | 1.017 |
| | FL | 508 | Natural-Log | Linear | ln(HtC) = 0.23*FL + 1.17 | 0.841 | -2066.9 | 0.131 |
| | AC | 508 | None | Linear | HtC = 0.39*AC—0.37 | 0.929 | -198.6 | 0.821 |
| | HdC | 508 | None | Linear | HtC = 0.51*HdC—2.28 | 0.929 | -198.6 | 0.821 |
| Heart Area | EGA* | 575 | None | Linear | HA = 0.65*EGA—9.22 | 0.972 | 287.1 | 1.281 |
| | BPD | 508 | Natural-Log | Linear | ln(HA) = 0.36*BPD—0.49 | 0.878 | -1500.9 | 0.228 |
| | FL | 508 | Natural-Log | Linear | ln(HA) = 0.46*FL + 0.59 | 0.851 | -1400.8 | 0.251 |
| | AC | 508 | Natural-Log | Linear | ln(HA) = 0.20*AC—0.69 | 0.903 | -1617.0 | 0.203 |
| | HdC | 508 | None | Linear | HA = 1.42*HdC—1.18 | 0.889 | 395.0 | 1.472 |
| Chest Circumference | EGA* | 575 | None | Linear | CC = 0.81*EGA—3.12 | 0.964 | 111.1 | 1.100 |
| | BPD | 508 | None | Linear | CC = 2.89*BPD + 8.09 | 0.927 | 314.4 | 1.360 |
| | FL | 508 | None | Linear | CC = 3.71*FL + 8.10 | 0.901 | 468.2 | 1.582 |
| | AC | 508 | None | Linear | CC = 1.63*AC + 8.87 | 0.952 | 101.5 | 1.103 |
| | HdC | 508 | None | Linear | CC = 1.67*HdC + 7.61 | 0.951 | 121.1 | 1.124 |
| Left Atrium | EGA | 534 | None | Linear | LA = 0.04*EGA—0.30 | 0.849 | -2268.1 | 0.119 |
| | BPD | 508 | None | Linear | LA = 0.15*BPD + 0.30 | 0.789 | -2011.2 | 0.138 |
| | FL | 508 | None | Linear | LA = 0.20*FL + 0.29 | 0.795 | -2025.3 | 0.136 |
| | AC | 508 | None | Linear | LA = 0.09*AC + 0.33 | 0.841 | -2153.7 | 0.120 |
| | HdC* | 508 | Natural-Log | Linear | ln(LA) = 0.05*HdC—1.59 | 0.858 | -2046.3 | 0.133 |
| Right Atrium | EGA | 534 | None | Linear | RA = 0.05*EGA—0.38 | 0.875 | -2200.5 | 0.127 |
| | BPD | 508 | None | Linear | RA = 0.19*BPD + 0.33 | 0.841 | -2000.2 | 0.139 |
| | FL | 508 | None | Linear | RA = 0.24*FL + 0.33 | 0.828 | -1959.9 | 0.145 |
| | AC | 508 | None | Linear | RA = 0.10*AC + 0.38 | 0.866 | -2089.0 | 0.128 |
| | HdC* | 508 | None | Linear | RA = 0.11*HdC + 0.29 | 0.878 | -2126.8 | 0.122 |
| Left Ventricle | EGA* | 534 | None | Linear | LV = 0.04*EGA—0.23 | 0.848 | -2381.6 | 0.107 |
| | BPD | 508 | None | Linear | LV = 0.14*BPD + 0.31 | 0.783 | -2114.3 | 0.125 |
| | FL | 508 | None | Linear | LV = 0.18*FL + 0.30 | 0.795 | -2133.8 | 0.121 |
| | AC | 508 | None | Linear | LV = 0.08*AC + 0.35 | 0.828 | -2231.4 | 0.111 |
| | HdC | 508 | None | Linear | LV = 0.08*HdC + 0.29 | 0.817 | -2200.0 | 0.114 |
| Right Ventricle | EGA | 534 | None | Linear | RV = 0.04*EGA—0.29 | 0.884 | -2414.9 | 0.104 |
| | BPD | 508 | None | Linear | RV = 0.16*BPD + 0.32 | 0.841 | -2160.6 | 0.118 |
| | FL | 508 | None | Linear | RV = 0.21*FL + 0.32 | 0.834 | -2137.7 | 0.121 |
| | AC | 508 | Natural-Log | Linear | ln(RV) = 0.10*AC—0.79 | 0.857 | -2045.6 | 0.133 |
| | HdC* | 508 | Natural-Log | Linear | ln(RV) = 0.11*HdC—0.89 | 0.889 | -2173.1 | 0.118 |
| Aortic Annulus | EGA* | 494 | None | Linear | Ao = 0.02*EGA—0.13 | 0.859 | -2843.9 | 0.056 |
| | BPD | 482 | None | Linear | Ao = 0.08*BPD + 0.19 | 0.811 | -2641.0 | 0.064 |

*(Continued)*

**Table 1.** (Continued)

| Structures[a] | Markers | n | Transformed | Best Fit Model | Best Fitting Equation | Adj.-R$^2$ | AIC | RMSE |
|---|---|---|---|---|---|---|---|---|
| | FL | 482 | Natural-Log | Linear | ln(Ao) = 0.22*FL—1.35 | 0.804 | -1929.9 | 0.135 |
| | AC | 482 | None | Linear | Ao = 0.04*AC + 0.21 | 0.841 | -2723.3 | 0.059 |
| | HdC | 482 | None | Linear | Ao = 0.05*HdC + 0.17 | 0.828 | -2688.3 | 0.061 |
| Pulmonary Annulus | EGA | 494 | Natural-Log | Linear | ln(PA) = 0.04*EGA—1.81 | 0.782 | -1937.0 | 0.140 |
| | BPD | 482 | None | Linear | PA = 0.08*BPD + 0.26 | 0.796 | -2539.4 | 0.072 |
| | FL | 482 | None | Linear | PA = 0.11*FL + 0.26 | 0.801 | -2552.4 | 0.071 |
| | AC | 482 | None | Linear | PA = 0.05*AC + 0.28 | 0.818 | -2594.8 | 0.068 |
| | HdC* | 482 | None | Linear | PA = 0.05*HdC + 0.22 | 0.829 | -2626.3 | 0.065 |
| Transverse Aortic Isthmus | EGA* | 494 | None | Quadratic | AI = -0.18*EGA2 + 1.97*EGA + 0.28 | 0.749 | -3239.1 | 0.038 |
| | BPD | 482 | None | Linear | AI = 0.03*BPD + 0.15 | 0.711 | -3102.8 | 0.040 |
| | FL | 482 | Inverse | Linear | 1/AI = -0.64*FL + 5.37 | 0.681 | -585.9 | 0.543 |
| | AC | 482 | Squared | Linear | AI2 = 0.01*AC + 0.01 | 0.709 | -3559.9 | 0.025 |
| | HdC | 482 | None | Linear | AI = 0.02*HdC + 0.14 | 0.739 | -3154.5 | 0.038 |
| Transverse Ductus Arteriosus | EGA | 494 | Natural-Log | Linear | ln(DA) = 0.03*EGA—2.16 | 0.674 | -1914.3 | 0.002 |
| | BPD | 482 | None | Linear | DA = 0.03*BPD + 0.17 | 0.659 | -3038.5 | 0.043 |
| | FL | 482 | None | Quadratic | DA = -0.13*FL2 + 1.31*FL + 0.32 | 0.678 | -3064.0 | 0.041 |
| | AC | 482 | None | Linear | DA = 0.02*AC + 0.18 | 0.679 | -3067.5 | 0.041 |
| | HdC* | 482 | None | Linear | DA = 0.02*HdC + 0.16 | 0.685 | -3076.7 | 0.041 |

*Forward stepwise selection criteria for 0.01 for model selection was utilized.

a. RMSE = root mean squared error;

b. AIC = Akaike's Information Criteria

†EGA = estimated gestational age, BPD = bi-parietal distance, FL = femur length, AC = abdominal circumference, HC = head circumference

trimester, with fewer cases in each third-trimester gestational weeks (n<10). The under-representation in later EGA of past studies, may have produced models that were under-sampled at later developmental stages resulting in higher order best-fitting equations that were not necessarily the most suitable models for "normal" development. Our data provides a balanced gestational sample that can provide more accurate summary throughout all cardiac gestational development ages.

When comparing correlation of fetal heart growth to other developmental markers, each fetal heart measurement is generally correlated with estimated gestational age (EGA). In detail, gross heart size (HW, HL, HtC, HA, and CC), LV, Ao, AI were best correlated to estimated gestational age, while LA, RA, RV, PA, and DA appeared to be better correlated with HdC. In fetal circulation, the majority of the cardiac output is carried out by right ventricle, while left ventricular output supplies blood flow to fetal brain [26]. Thus, left heart structures may theoretically be better correlated with fetal head growth. However, our data suggests the opposite. This paradoxical finding implies that head growth is not solely affected by size of left heart. In summary, fetal heart growth is generally well-correlated with gestational age or head circumferences. For certain fetal heart structures, head circumferences can be used as a developmental marker to aid in predicting fetal heart growth.

A review of recent cardiac developmental nomograms providing guidance on developing nomograms indicates that cardiac development in fetuses has been shown to vary between races[27], indicating the need for developing accurate centiles and nomograms that reflect Asian cardiac development. When comparing the centile graphs of RV and LV by EGA (See Supplementary Materials, S1h & S1i Fig) to those of Shapiro et al[3] from Israel, and

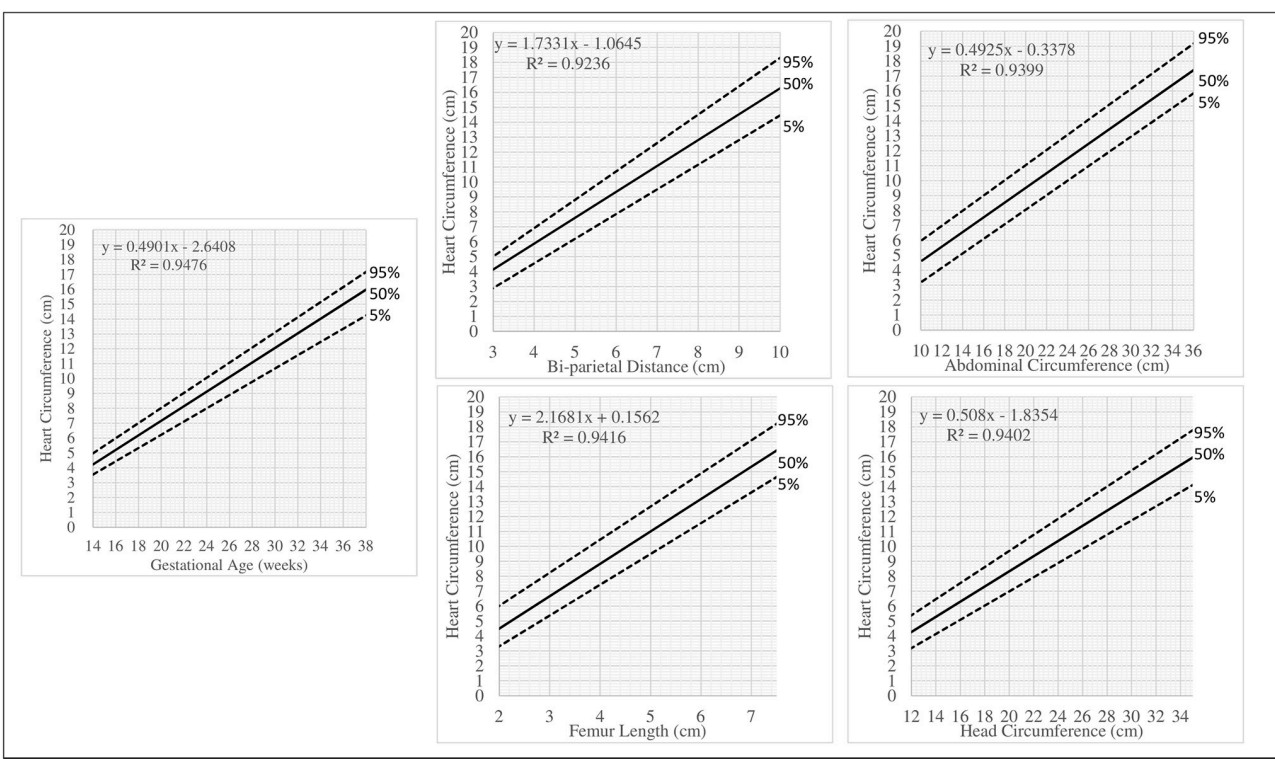

**Fig 2. Centile graphs for heart circumference by estimated gestational age, bi-parietal distance, femur length, abdominal circumference, head circumference.**

Gabbay-Benziv et al[28] from the United States, our best fit lines were both linear, while RV and LV were higher order equations in both the other studies. Despite this difference, the range of development by EGA followed a similar trend to ours in earlier development, however the range of normality tended to be wider at later stages of development. The mean width for LV was slightly lower, for example, at 33 weeks gestation, the mean LV dimensions was 1.09cm compared to 1.36cm from the American population and 1.15cm in the Israeli population. We produced centile graphs and nomograms that were similar to the American study by Krishnan et al [10]and the Canadian study by Schneider et al[6] for Ao, PA by EGA, BPD and FL. Our normal ranges (See Supplementary Materials, S1j & S1k Fig) had a similar trend for Ao and PA by BPD, with a slightly lower range of normality at earlier ages, but a higher rate of development at later developmental stages. The difference in development pattern in our sample may suggest the need for consideration of race when comparing fetal cardiac development.

Compared to nomogram z-score calculations from previous fetal cardiac nomogram studies [6–8, 29, 30], using the same parameters reported by Cantinotti et al (*Developmental markers*: EGA = 28 week, FL = 5.2 cm, BPD = 7.2 cm and Ao = 0.35cm, AI = 0.2)[27], our nomograms produced the following z-scores for Ao (GA: z = -2.30, FL: z = -3.42, and BPD: z = -2.42). Our calculations for Ao fell mid-range compared to the calculations by nomograms from previous studies (Ranges: GA: -3.97 ~ -1.83; FL: -4.04 ~ -1.1; BPD: -3.77 ~ -1.58), and were further from normal development for EGA, FL, and BPD, compared to Krishnan et al., Schneider et al. and McElhinney et al., but were closer to normal development than Lee et al. and Pasquini et al. Moreover, we produced nomograms with the same methodology and parameter (Ao*FL, LV*FL, PA*FL, RV*FL, Ao*GA, PA*GA) as Schneider et al [6] as well a

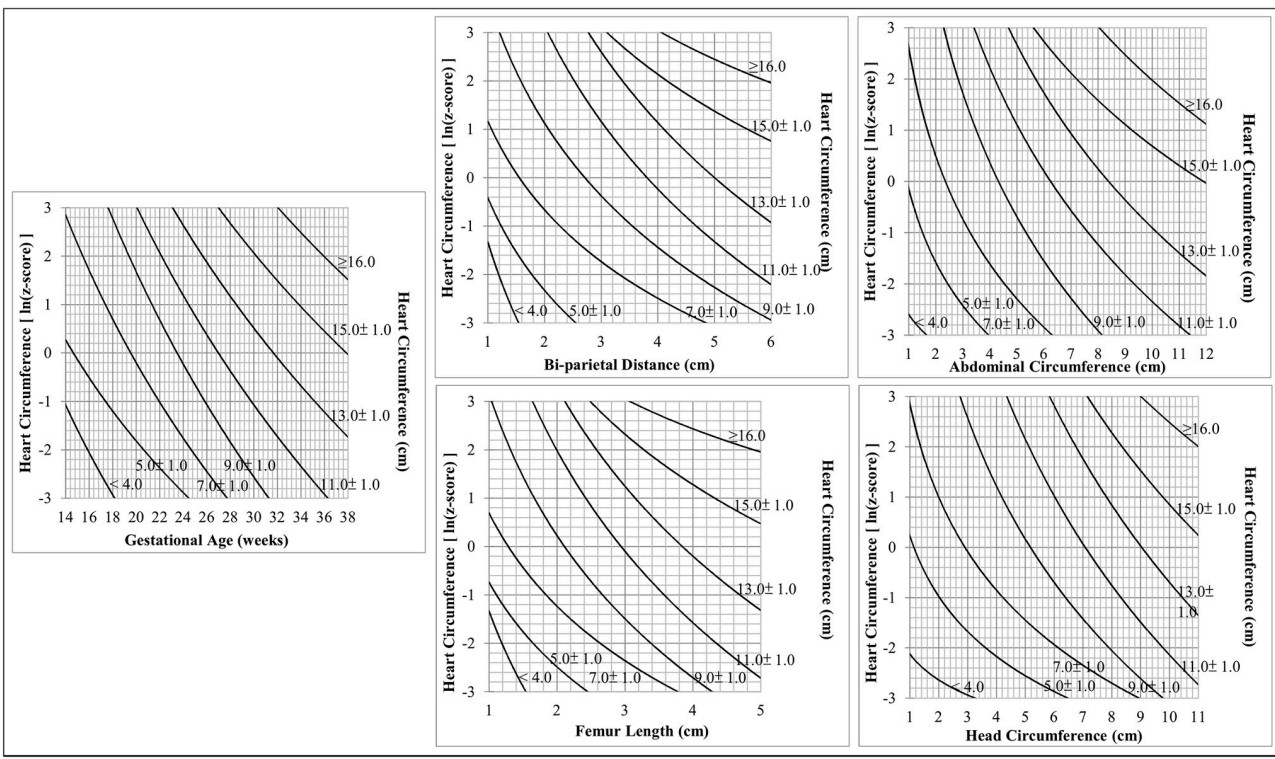

**Fig 3. Nomogram for heart circumference by estimated gestational age, bi-parietal distance, femur length, abdominal circumference, head circumference.**

variety of other parameters that were not included. Although our nomograms followed a similar trend in development, the normal growth curves were shifted left on the x-axis, indicating that development was occurring at a slower rate in our sample than in the Caucasian sample. We hope to share our best fitting equations and nomograms online on mobile apps and websites that measure fetal echocardiography development (eg. parameterz.com, BabyNorm, etc.) [31], to supplement previously developed nomograms and provide novel nomograms for parameters that have not yet been reported. Our measurements could be easily accessible to both patients and physicians alike who are need to compare their measurements among an Asian sample.

## Limitations

There are a few possible limitations. Developmental normality was determined during the neonatal period and thus some genetic syndromes or chromosomal abnormalities may have been missed during the neonatal stage. First, although our sample was more evenly distributed throughout the gestational period, our sample is relatively small compared to some previous Caucasian studies [8, 12]. A further larger scale study to validate current finding may be necessary in a Taiwanese sample. Second, measurements may have been influenced by intra-observer bias, since only one ultrasound observer collected data. Despite this limitation the observer was an experienced operator, and therefore measurement errors were less likely to be present, however interpretation of the findings should be kept in mind, as the reference ranges likely did not account for inexperienced operator error, as well as failing to capture inter-

observer variability. Third, our sample may be confounded by the fact that sampling was done from an unselected and non-randomized population, participants attending the 3 clinics may have confounding factors that were not accounted for and thus may have influenced the reference ranges. Fourth, our sample include cases conceived by assisted reproductive technologies (ART). The use of ART may have an impact on the fetal heart, although the mechanism may be confounded by intrauterine growth restriction and factors related to causes of infertility [32]. As we prospectively exclude cases with growth restriction, the proportion of ART cases in our sample were 4%, which was similar to general population in Taiwan [33]. Despite concerns about the effect of ART, our sample may represent the heterogenicity of fetal heart growth in Asian fetus without growth restriction. Lastly, some helpful measurements are not included, for example, ventricular thickness, diameters of bilateral peripheral pulmonary arteries and diameter of aortic isthmus in sagittal view.

## Conclusions

The challenge of prenatally diagnosing congenital heart disease is not to diagnose the condition itself, but rather to predict the fetal or post-natal outcomes based on reference ranges and to select cases that may benefit from fetal intervention, where available. Nomograms are practical to use in clinical practice for quick and manual calculations of z-scores for guiding clinical decisions, which is not yet sufficiently established for fetal development in an Asian population. Since there is significant geographical differences in the birth prevalence of CHD worldwide, using reference ranges developed from specific racial populations would be more suitable in confirming normal fetal cardiac development.

## Supporting information

**S1 Fig.** a-m. Centile graphs by estimated gestational age, bi-parietal distance, femur length, abdominal circumference, head circumference.
(PDF)

**S2 Fig.** a-m. Nomogram for estimated gestational age, bi-parietal distance, femur length, abdominal circumference, head circumference.
(PDF)

**S1 Data.**
(XLSX)

## Acknowledgments

Assistance in statistical computation and visualization was provided by Chan-Yu Sung, a research fellow from Taiji Clinic was greatly appreciated.

## Author Contributions

**Conceptualization:** Shu-Jen Yeh, Shan-Miao Lin, Yu-Ching Chou, Ming-Ren Chen, Tung-Yao Chang.

**Formal analysis:** Eric C. Lussier.

**Investigation:** Yu-Ching Chou, Szu-Ping Huang, Ming-Ren Chen.

**Methodology:** Eric C. Lussier, Yu-Ching Chou, Ming-Ren Chen.

**Project administration:** Yu-Ching Chou, Szu-Ping Huang, Tung-Yao Chang.

**Resources:** Ming-Ren Chen, Tung-Yao Chang.

**Supervision:** Yu-Ching Chou, Ming-Ren Chen, Tung-Yao Chang.

**Validation:** Tung-Yao Chang.

**Writing – original draft:** Eric C. Lussier, Wan-Ling Chih.

**Writing – review & editing:** Eric C. Lussier, Shu-Jen Yeh, Wan-Ling Chih, Shan-Miao Lin.

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
