## [Decision Letter · Decision Letter 0]

10 Mar 2020

PONE-D-20-03811

Reference ranges and Z-scores for fetal cardiac measurements from two-dimensional echocardiography in Asian population

PLOS ONE

Dear Dr. Chen,

Thank you for submitting your manuscript to PLOS ONE. After careful consideration, we feel that it has merit but does not fully meet PLOS ONE’s publication criteria as it currently stands. Therefore, we invite you to submit a revised version of the manuscript that addresses the points raised during the review process.

Please address all the issue raised by the reviewers before proceeding with the resubmission.

We would appreciate receiving your revised manuscript by Apr 24 2020 11:59PM. To enhance the reproducibility of your results, we recommend that if applicable you deposit your laboratory protocols in protocols.io, where a protocol can be assigned its own identifier (DOI) such that it can be cited independently in the future. For instructions see: http://journals.plos.org/plosone/s/submission-guidelines#loc-laboratory-protocols

We look forward to receiving your revised manuscript.

Kind regards,

Elena Cavarretta, M.D., Ph.D.

Academic Editor

PLOS ONE

Journal Requirements:

Reviewers' comments:

Reviewer's Responses to Questions

**Comments to the Author**

1. Is the manuscript technically sound, and do the data support the conclusions?

Reviewer #1: Yes

Reviewer #2: Partly

2. Has the statistical analysis been performed appropriately and rigorously? 

Reviewer #1: Yes

Reviewer #2: Yes

3. Have the authors made all data underlying the findings in their manuscript fully available?

Reviewer #1: Yes

Reviewer #2: Yes

4. Is the manuscript presented in an intelligible fashion and written in standard English?

Reviewer #1: Yes

Reviewer #2: Yes

5. Review Comments to the Author

Reviewer #1: The article is nice, well written, good statistical analysis

There a few points that merit attention

1) In the method sections and in the tables is not really clear how measurements are made and what they represent (eg. Left atrium, right atrium, right ventricle are not measurements, specify what you measure and insert units, millimeter, cmq or what else). Please specify well. I woul add a table on how measurements have been performed

2) A comparsion with other nomograms would be very important. Are the range of normality higher or lower than those previously proposed mainly based on Caucasian population?

In this context discuss and cite also

Cantinotti et al. Limitations of Current Fetal Echocardiography Nomograms for 2D Measures: A Critical Overview and Analysis for Future Research. J Am Soc Echocardiogr. 2018 Dec;31(12):1368-1372.e10.

3) These nomograms may be incorporated in current web site providng nomograms (e. parameterz) and mobile App (such as BabyNorm). Please comment and cite at this aim

Cantinotti et al. Pediatric echocardiographic nomograms: What has been done and what still needs to be done.Trends Cardiovasc Med. 2017 Jul;27(5):336-349.

4) some helpful measuremensta are missing such as ventricular thickness. Please add in the limitations

5) Limitations and Conclusive remarks should be separate by discussion, please add subtitles

Reviewer #2: This is a well-designed prospective study done in a low risk population including almost 600 cases to define normal ranges (Z-scores) of different echocardiographic variables.

The strengths of the paper include:

- reasonable sample size across the range of gestational age.

- clearly defined methodology.

- The authors have used both gestational age and different fetal biometries to base their normal ranges on.

-Extensive documentation is provided with graphs representing 5th and 95th percentiles

However, several issues should be adressed:

Abstract: results section in the abstract should be reduced

Methodology:

• No baseline characteristics are included such as use of assisted reproductive techniques or maternal diabetes which may have an impact on the fetal heart (Valenzuela-Alcaraz B, Circulation 2013; Patey O, UOG 2019).

• No comprehensive fetal assessment was performed including estimated fetal weight assessment. Fetal growth restricted fetuses (under the 10th percentile) shouldn’t be included in nomograms’ construction as they may present cardiac remodelling (Rodríguez-López M, UOG 2017).

• Atrial measurement should be performed at its maximal distension (end-systole) JS Carvalho UOG 2013.

• No inter or intraobserver variability was studied.

• There is no data available on the feasilibity of the studied parameters.

Discussion

• Should be reduced.

• There is no consistent data to state that “For cases with uncertain gestational age, head circumference could be used as an optimal developmental marker to predict fetal heart growth” in line 347-349

6. PLOS authors have the option to publish the peer review history of their article (what does this mean?). If published, this will include your full peer review and any attached files.

Reviewer #1: Yes: Massimiliano Cantinotti

Reviewer #2: No

---

## [Author Response · Author response to Decision Letter 0]

22 Apr 2020

Responses to the Reviewers

Reviewer #1: The article is nice, well written, good statistical analysis. There a few points that merit attention:

1) In the method sections and in the tables is not really clear how measurements are made and what they represent (eg. Left atrium, right atrium, right ventricle are not measurements, specify what you measure and insert units, millimeter, cmq or what else). Please specify well. I would add a table on how measurements have been performed.

Thank you for the suggestion. We have specified how the measurement was made more clearly in Lines 80-95. Units of measurements were explicitly stated as using cm and cm2. In summary, we measured heart & chest size, chamber width, aortic annulus (Ao) and pulmonary annulus (PA) in diastole, while measured transverse aortic isthmus (AI) and transverse ductus arteriosus (DA) in its widest systolic diameter. We chose to measure Ao and PA annulus in diastole but not in systole, in order to optimize measurement especially in earlier gestational age. Our measurement dimensions were established based on past studies measuring Ao and PA annulus during diastole (Sharland et al., 1992; Shapiro et al., 1998; Trisha V. Vigneswaran et al., 2018).

2) A comparison with other nomograms would be very important. Are the range of normality higher or lower than those previously proposed mainly based on Caucasian population?

In this context discuss and cite also Cantinotti et al. Limitations of Current Fetal Echocardiography Nomograms for 2D Measures: A Critical Overview and Analysis for Future Research. J Am Soc Echocardiogr. 2018 Dec;31(12):1368-1372.e10.

Thank you for providing this suggestion and references for support. We have added two new paragraph that add to the discussion section with a comparison between centile graphs and nomograms from our findings to this paper and other previous studies (Lines 247-280).

3) These nomograms may be incorporated in current web site providng nomograms (e. parameterz) and mobile App (such as BabyNorm). Please comment and cite at this aim.

Cantinotti et al. Pediatric echocardiographic nomograms: What has been done and what still needs to be done.Trends Cardiovasc Med. 2017 Jul;27(5):336-349.

Thank you for sharing these resources. We have included a section in the conclusion and added the following text to our discussion (Lines 280-286):

“We hope to share our best fitting equations and nomograms online on mobile apps and websites that measure fetal echocardiography development (eg. parameterz.com, BabyNorm, etc.)(Cantinotti et al, 2017), to supplement previously developed nomograms and provide novel nomograms for parameters that have not yet been reported. Our measurements could be easily accessible to both patients and physicians alike who are need to compare their measurements among an Asian sample”

4) Some helpful measurements are missing such as ventricular thickness. Please add in the limitations.

We have added the missing parameters as limitations in our study (Lines 310-312).

5) Limitations and Conclusive remarks should be separate by discussion, please add subtitles

Thank you to the reviewer for this point. We have added subheadings for Limitation and Conclusion section (Line 288 & Line 314).

Reviewer #2: This is a well-designed prospective study done in a low risk population including almost 600 cases to define normal ranges (Z-scores) of different echocardiographic variables.

The strengths of the paper include:

- reasonable sample size across the range of gestational age.

- clearly defined methodology.

- The authors have used both gestational age and different fetal biometries to base their normal ranges on.

-Extensive documentation is provided with graphs representing 5th and 95th percentiles

However, several issues should be addressed:

Abstract: 

Results section in the abstract should be reduced

Thank you to the reviewer for his feedback on the abstract. We have shortened the abstract, especially the results section.

Methodology:

• No baseline characteristics are included such as use of assisted reproductive techniques or maternal diabetes which may have an impact on the fetal heart (Valenzuela-Alcaraz B, Circulation 2013; Patey O, UOG 2019).

We retrospectively excluded cases with any maternal disease diagnosed during pregnancy, including GDM and cases of growth restriction in later pregnancy or small for gestational age at birth. The proportion of ART cases in our sample is 4%, which is similar to general population in Taiwan (4.3%) (Hsu JC, Su YC, Tang BY, Lu CY). Use of assisted reproductive technologies before and after the Artificial Reproduction Act in Taiwan. PloS one. 2018;13(11):e0206208). We believe our sample may represent the heterogenicity of fetal heart growth in Asian fetus without growth restriction. However, for clarity we have added a new paragraph in the limitation section illustrating the issue of including ART cases in our sample (Lines 303-310).

• No comprehensive fetal assessment was performed including estimated fetal weight assessment. Fetal growth restricted fetuses (under the 10th percentile) shouldn’t be included in nomograms’ construction as they may present cardiac remodelling (Rodríguez-López M, UOG 2017).

We prospectively excluded any fetuses that had FGR so they would not have impacted our sample, and only represented normally developed fetuses. We have clarified the inclusion & exclusion criteria in method section and added reference of the fetal biometry we used (Line 56-57).

• Atrial measurement should be performed at its maximal distension (end-systole) JS Carvalho UOG 2013.

Thank you for feedback on this point. Atrial size in end-systole represents the largest diameter of atrium in cardiac cycle. However, we chose to measure in diastole, which is the same as Shapiro et al., 1998, and also hope to provide a reference with simple measuring process. 

No inter or intraobserver variability was studied.

This is indeed a limitation of our study, since we only had one sonographer, interobserver variability was not studied. However, to ensure quality of measurement and assessment, all measurements were confirmed by an obstetrician-gynecologist and a pediatric cardiologist during patient visits. We have explained this in both method and limitation sections (Line 76-78 & line 295-300).

• There is no data available on the feasibility of the studied parameters.

Thank you for the feedback. We have added reference 14-22 in the introduction section to justify the description about feasibility of the studied parameters. (Line 39-42):” In clinical practice, z-scores references are practical not only in the screening and diagnosis of fetal cardiac structural abnormalities (reference 14-18), but fetal cardiologist also use z-scores to predict and counsel about possible postnatal outcome and treatment strategies (reference 19-22).”

• Discussion should be reduced.

Thank you for your feedback on the discussion section. We have shortened the discussion section for parts that were repeating and have rearranged some sections for clarity and conciseness. 

• There is no consistent data to state that “For cases with uncertain gestational age, head circumference could be used as an optimal developmental marker to predict fetal heart growth” in line 347-349

We have revised this statement to now state, “For certain fetal heart structures head circumferences can be used as a developmental marker to aid in predicting fetal heart growth.” (Line 245-246)

---

## [Decision Letter · Decision Letter 1]

30 Apr 2020

Reference ranges and Z-scores for fetal cardiac measurements from two-dimensional echocardiography in Asian population

PONE-D-20-03811R1

Dear Dr. Chen,

We are pleased to inform you that your manuscript has been judged scientifically suitable for publication and will be formally accepted for publication once it complies with all outstanding technical requirements.

With kind regards,

Elena Cavarretta, M.D., Ph.D.

Academic Editor

PLOS ONE

Additional Editor Comments (optional):

Reviewers' comments:

Reviewer's Responses to Questions

**Comments to the Author**

1. If the authors have adequately addressed your comments raised in a previous round of review and you feel that this manuscript is now acceptable for publication, you may indicate that here to bypass the “Comments to the Author” section, enter your conflict of interest statement in the “Confidential to Editor” section, and submit your "Accept" recommendation.

Reviewer #1: All comments have been addressed

2. Is the manuscript technically sound, and do the data support the conclusions?

Reviewer #1: Yes

3. Has the statistical analysis been performed appropriately and rigorously? 

Reviewer #1: Yes

4. Have the authors made all data underlying the findings in their manuscript fully available?

Reviewer #1: Yes

5. Is the manuscript presented in an intelligible fashion and written in standard English?

Reviewer #1: Yes

6. Review Comments to the Author

Reviewer #1: Authors correctely addressed all points raised by the reviewers. Good job! The artice is nice, and provide interesting data that may improve current fetal echocardiographic nomograms

7. PLOS authors have the option to publish the peer review history of their article (what does this mean?). If published, this will include your full peer review and any attached files.

Reviewer #1: Yes: Massimiliano Cantinotti M.D.

---

## [Editor Report · Acceptance letter]

10 Jun 2020

PONE-D-20-03811R1 

Reference ranges and Z-scores for fetal cardiac measurements from two-dimensional echocardiography in Asian population 

Dear Dr. Chen:

I'm pleased to inform you that your manuscript has been deemed suitable for publication in PLOS ONE. Congratulations! Your manuscript is now with our production department. 

Kind regards, 

on behalf of

Dr. Elena Cavarretta 

Academic Editor

PLOS ONE